# Seasonal antigenic prediction of influenza A H3N2 using machine learning

Syed Awais W. Shah [1], Daniel P. Palomar [1,2], Ian Barr [3,4], Leo L. M. Poon [5,6], Ahmed Abdul Quadeer [1,7] ✉ & Matthew R. McKay [4,7] ✉

Antigenic characterization of circulating influenza A virus (IAV) isolates is routinely assessed by using the hemagglutination inhibition (HI) assays for surveillance purposes. It is also used to determine the need for annual influenza vaccine updates as well as for pandemic preparedness. Performing antigenic characterization of IAV on a global scale is confronted with high costs, animal availability, and other practical challenges. Here we present a machine learning model that accurately predicts (normalized) outputs of HI assays involving circulating human IAV H3N2 viruses, using their hemagglutinin subunit 1 (HA1) sequences and associated metadata. Each season, the model learns an updated nonlinear mapping of genetic to antigenic changes using data from past seasons only. The model accurately distinguishes antigenic variants from non-variants and adaptively characterizes seasonal dynamics of HA1 sites having the strongest influence on antigenic change. Antigenic predictions produced by the model can aid influenza surveillance, public health management, and vaccine strain selection activities.

Genetic changes accumulated in the influenza virus population may alter their antigenic properties, resulting in antigenic drift[1]. Antigenically drifted influenza strains may escape immunity induced by previous infection or vaccination[2], leading to an increase in morbidity and mortality[1]. To counter antigenic drift, influenza virus strains included in the human influenza vaccine are regularly updated. The World Health Organization (WHO) holds vaccine composition meetings (VCMs) twice each year to recommend vaccine strains for the upcoming northern hemisphere (NH) and southern hemisphere (SH) influenza seasons[3]. Genetic and antigenic characteristics of circulating isolates are considered when recommending vaccine strains at each meeting[3].

Antigenic characteristics of circulating isolates are primarily determined through hemagglutination inhibition (HI) assays utilizing ferret post-infection antisera, although assessments using both human

pre- and post-vaccination antisera are also conducted[3]. The HI assay measures the cross-reactivity of a test virus isolate to an antiserum raised against a reference virus isolate in ferrets or against the vaccine viruses in humans. Ferret antisera are produced in naïve animals and hence have high specificity compared to human antisera which generally have extensive cross-reactive antibodies due to encountering multiple infections or vaccinations against influenza. Large-scale antigenic characterization of circulating isolates using HI assays incurs high costs and is time and labor-intensive[4,5].

Computational methods that predict ferret HI titers of influenza viruses using genetic sequence data may help to reduce these burdens[1,6]. Accurate sequence-based models could enable more comprehensive antigenic surveillance of circulating virus isolates without the need for increased experimental resources[1]. The efficiency of evolutionary monitoring and vaccine selection procedures may be

[1]Department of Electronic and Computer Engineering, The Hong Kong University of Science and Technology, Clear Water Bay, Hong Kong SAR, China. [2]Department of Industrial Engineering & Decision Analytics, The Hong Kong University of Science and Technology, Clear Water Bay, Hong Kong SAR, China. [3]WHO Collaborating Centre for Reference and Research on Influenza, Melbourne, Victoria, Australia. [4]Department of Microbiology and Immunology, University of Melbourne, at The Peter Doherty Institute for Infection and Immunity, Melbourne, Victoria, Australia. [5]School of Public Health, LKS Faculty of Medicine, The University of Hong Kong, Hong Kong SAR, China. [6]Centre for Immunology & Infection, Hong Kong SAR, China. [7]Department of Electrical and Electronic Engineering, University of Melbourne, Melbourne, Victoria, Australia. ✉e-mail: ahmed.quadeer@unimelb.edu.au; matthew.mckay@unimelb.edu.au

improved by providing targeted sets of isolates for experimental evaluation. Furthermore, by learning the complex map from genetic to antigenic changes, accurate prediction methods could yield new insights into influenza evolution and the processes underpinning antigenic drift[2,7].

Here we develop a machine learning (ML) model that predicts antigenic properties of influenza A virus (IAV) H3N2 isolates circulating in a season using their HA1 sequences and associated metadata while being trained on data from past seasons only. The model is designed and evaluated for predicting, on a season-by-season basis, HI titers of virus-antiserum pairs involving viruses sequenced globally as part of WHO's seasonal influenza surveillance. This approach is distinct from previous sequence-based HI titer prediction methods[4,7–11], which in many cases have considered the problem of predicting HI titers of virus-antiserum pairs randomly selected over time. For training and testing our model, we use the IAV H3N2 antigenic data of influenza seasons from 2003 – 2021 reported by the Worldwide Influenza Center at the Francis Crick Institute[12], genetic data available at influenza sequence databases[13,14], and their associated metadata. The model predicts HI titers of virus-antiserum pairs with a mean absolute error (MAE) of 0.702 antigenic units (where 1 antigenic unit ≈ 2-fold change in HI titer) per season and exhibits a strong discriminatory ability in distinguishing antigenic variants across seasons. The developed data-driven ML model captures from data the nonlinear effects in the relation between IAV H3N2 antigenic and genetic changes, which has been suggested by recent experimental studies[15,16]. We show that the model's predictive power is robust to limiting training data per season. Moreover, incorporating a small amount of antigenic data from circulating isolates in model training significantly enhances its accuracy, particularly for seasons associated with strong antigenic drift. The model identifies key sites with the strongest impact on IAV antigenic change, most of which are located in HA1 epitopes, and reveals how they vary across different seasons. Overall, accurate prediction of HI titers by the developed model across seasons shows its viability for seasonal antigenic characterization of IAV H3N2.

## Results

### Machine learning model for seasonal antigenic characterization of IAV H3N2

Our ML model for seasonal antigenic characterization of IAV H3N2 was designed under a seasonal framework (Fig. 1a) that mimics the WHO VCM protocols[12] (Supplementary Fig. 1). The NH VCM is held each February and considers antigenic data for circulating isolates from the preceding September to January, while the SH VCM is held each September and considers isolates from the preceding February to August. Each of these periods constitutes an influenza season. Under the seasonal framework, for any given season, our model is trained using genetic, antigenic, and metadata information available prior to that season. The trained model predicts antigenic data for the current season based on genetic data of isolates circulating in that season, along with metadata (Fig. 1a).

The model employs an adaptive boosting method (AdaBoost)[17,18] consisting of an ensemble of decision trees (Fig. 1b). The model is trained in a supervised fashion and learns a nonlinear mapping from genetic difference to antigenic difference (defined as normalized HI titers (NHT); "Methods") between virus-antiserum pairs of past isolates. Pairwise genetic difference is based on the HA1 gene of isolates in a virus-antiserum pair and is encoded using the GIAG010101 mutation matrix from the amino acid index 2 (AAindex2) database[19] ("Methods", Fig. 1c). The model also utilizes metadata information including virus avidity[7], antiserum potency[7], and passage category (egg or cell) of virus isolates and antisera, which is represented using one-hot encoding ("Methods", Fig. 1c). The trained model predicts the antigenic differences (in terms of NHTs) of circulating virus isolates using only their HA1 genetic sequences and metadata information (Fig. 1b, c).

### Model training, optimization, and validation

For model training, optimization, and evaluation, we compiled HI titers data of IAV H3N2 from reports published by the Worldwide Influenza Center (WIC) at the Francis Crick Institute, London,[12] and genetic data from influenza sequence databases, GISAID[13] and IVR[14]. The processed dataset included NHTs of 36,709 virus-antiserum pairs with corresponding metadata, spanning 37 influenza seasons from 2003NH to 2021SH ("Methods"). Data availability was limited in the early seasons and increased progressively over time (Supplementary Fig. 2a). Preliminary assessment using a baseline model ("Methods") revealed sufficient data for reliable predictive performance from the 2012NH season onwards (Supplementary Fig. 2b). The four seasons 2012NH to 2013SH were selected as validation seasons to perform feature selection and model optimization.

Using the validation seasons, it was found that incorporating all four metadata features provided optimal performance (MAE of 1.091) (Supplementary Fig. 3a) and substantially outperformed the baseline model trained with no metadata (MAE of 1.641). The metadata captures distinct information: virus avidity and antiserum potency account for experimental variations among HI assays[7], while the passage category informs about antigenicity-altering mutations incurred during in vitro propagation of virus isolates using cell or egg lines[1,20]. Optimization of the model hyperparameters significantly improved performance (from MAE of 1.091 to 0.759) over validation seasons (Supplementary Fig. 3b). Selecting the optimal amino acid mutation matrix for genetic data encoding (Supplementary Fig. 3c; for details see "Methods") further slightly improved performance (MAE of 0.75).

### Optimized model accurately performs seasonal antigenic characterization

The performance of the optimized model in predicting NHTs was evaluated for each of the 14 test seasons (2014NH to 2020SH). This yielded a MAE, averaged across seasons, of 0.702 antigenic units (Fig. 2a). Predictions were generally more accurate in more recent influenza seasons, likely due to the increased availability of data over time (Supplementary Fig. 2a). Further experiments assessed the robustness of our model to variations in the training data. Prediction accuracy was retained even under conditions where there is substantially less antigenic data for training (Supplementary Fig. 4a, b). Minimal effects on performance (compromised performance for a single season only) were observed when omitting HI titers data from an entire season (Supplementary Fig. 4c).

The ability of our model to detect antigenic variants was also examined. An influenza virus is considered antigenically distinct from the virus used to generate the antiserum if a more than 4-fold reduction in HI titers is observed against the antiserum[4,21]. Our model classified antigenic variants and non-variants with an average area under the receiver operating characteristic (AUROC) of 92% across the 14 test seasons (Fig. 2b). Additional metrics (e.g., sensitivity and specificity) further demonstrated classification accuracy (Supplementary Fig. 5a). We have incorporated the model into a web application (https://huggingface.co/spaces/sawshah/SAP_H3N2) that reports predicted NHTs for user-specified H3N2 virus-antiserum pairs (see "Methods").

To further calibrate model performance, we assessed alternative approaches. These included a linear method (NextFlu substitution model[7]), ML methods (random forest (RF)[22] and extreme gradient boosting (XGBoost)[23]), and neural network methods (multi-layer perceptron (MLP)[24] and residual neural network (ResNet)[24]). These models, along with their implementation details, are described in "Methods". Among these alternative models, NextFlu is the most widely used model for antigenic prediction. It has been employed to predict NHTs under a non-seasonal framework, where the model was trained on data spanning all time periods and the predictions were made for randomly selected historical NHTs. When evaluated under the seasonal prediction framework (Fig. 1a) over 14 test seasons

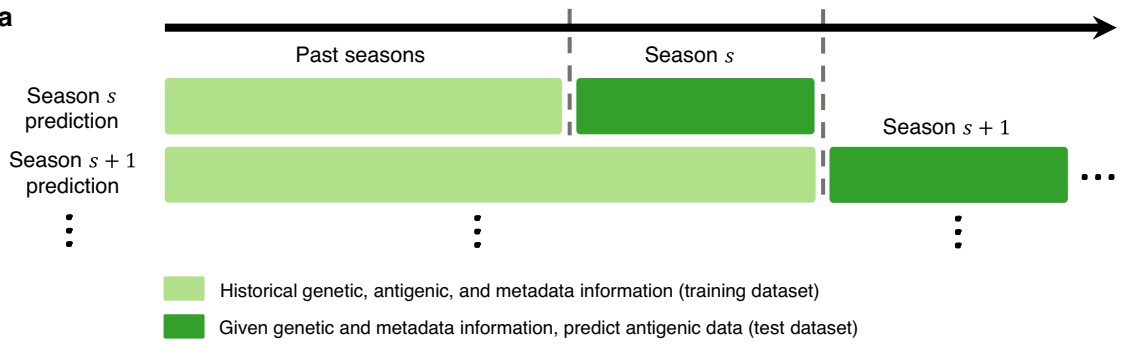

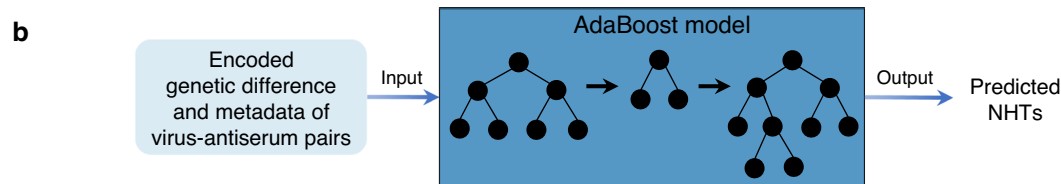

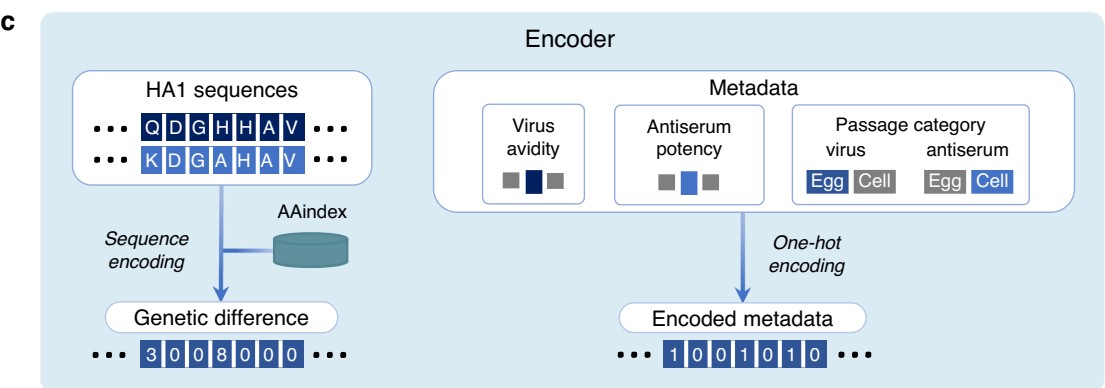

**Fig. 1 | Overview of the seasonal framework and the designed ML method for seasonal antigenic characterization of IAV H3N2. a** Seasonal division of data into training and test datasets respectively for training and evaluation of computational methods in a time-series fashion. Under this framework, historical genetic, antigenic, and metadata information of virus isolates from past seasons is included in the training dataset, while genetic and metadata information of virus isolates from the current season form the test dataset. **b** The trained AdaBoost model was used to predict NHTs using only encoded genetic difference and metadata information of virus-antiserum pairs. **c** Details of the encoding performed at the input of the AdaBoost model. The HA1 sequences of isolates in each virus-antiserum pair were encoded using the amino acid mutation matrix available in the AAindex2 database. One-hot encoding was used to represent the metadata information, which includes virus avidity, antiserum potency, and passage category of isolates. The encoded genetic difference and metadata of each virus-antiserum pair were used as input features of the AdaBoost model. Each training virus-antiserum pair was labeled by NHT-based antigenic difference (see "Methods").

(2014NH to 2020SH), the AdaBoost model achieved the best performance (MAE of 0.702), followed by the ML methods (MAE of 0.72 and 0.738 respectively for XGBoost and RF) and the NextFlu model (MAE of 0.819). The neural network methods achieved the worst performance (MAE of 0.964 and 0.986 respectively for ResNet and MLP) (Supplementary Fig. 6).

**Partial antigenic information of circulating isolates alleviates antigenic drift effects**

Disregarding the initial season of 2014, the MAE of our optimized model was well below average in two seasons: 2016NH and 2019NH (Fig. 2a). This appears to be attributed to a larger antigenic drift observed in these seasons, which is evident from the presence of circulating isolates (red circles) that are widely dispersed from isolates of past seasons (gray points) (Fig. 3a). Importantly, performance was recuperated in subsequent seasons and degradation was not carried forward (Fig. 2a).

While significant antigenic drift makes prediction more challenging, access to partial antigenic data for circulating virus isolates in a season may help overcome this challenge. Further analysis confirmed this hypothesis. For each test season, including as little as 10% of the antigenic data for circulating isolates in the model training improved performance uniformly, with the most significant gains observed in those seasons with large antigenic drift (Fig. 3b and Supplementary Fig. 7). Access to a small amount of antigenic data can therefore help ensure high prediction accuracy irrespective of the level of drift experienced by IAV H3N2.

**Antigenically important sites identified by the model are temporally associated with HA1 epitopes**

Analysis of historical data has demonstrated that the antigenic evolution of influenza is strongly influenced by mutations at a subset of sites within HA1[7,25,26]. Our model enables the identification of the specific

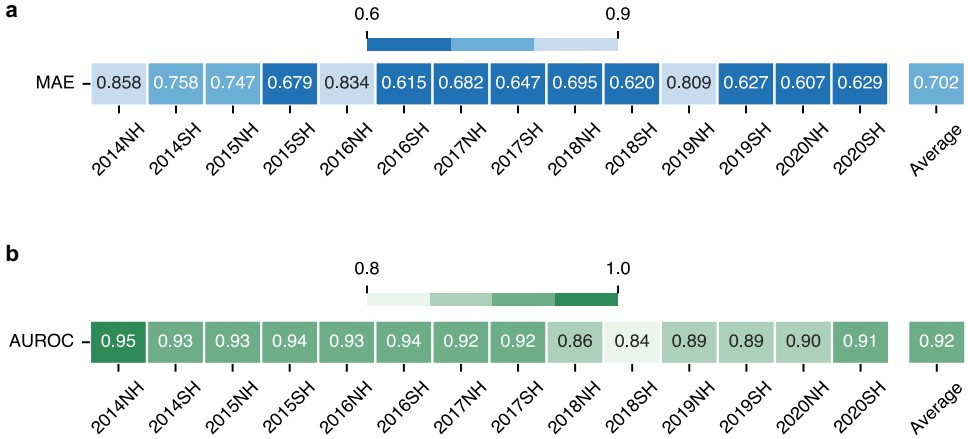

**Fig. 2 | Performance of the optimized model for seasonal antigenic characterization of IAV H3N2. a, b** Model prediction and classification performance is shown in terms of (**a**) MAE and (**b**) AUROC respectively over 14 test seasons from 2014NH to 2020SH. The optimized model consisted of encoded genetic difference using the best-performing amino acid mutation matrix GIAG010101, optimized hyperparameters (see "Methods"), and all features in the metadata information (virus avidity, antiserum potency, and passage category (egg or cell) of virus

isolates and antisera) (Supplementary Fig. 3). The classification score AUROC was obtained by converting the measured and predicted NHTs to binary labels such that if NHT was greater than 2 units it was assigned a binary label 1, otherwise 0. The 'Average' cell in (**a, b**) indicates the score averaged over 14 test seasons from 2014NH to 2020SH. The darker color cells indicate better performance. Source data are provided as a Source Data file.

HA1 sites that have the greatest effect on antigenic changes during a given season, providing insights into the seasonal dynamics of these sites. Such sites can be predicted based on their feature importance[22] scores from the model ("Methods"). Aggregating the top 20 sites identified for seasons 2014NH to 2020SH revealed 30 important sites in total (Fig. 4a). Of these, 25 were located within established HA1 epitopes[25,27,28] (A, B, C, D, and E). Substitutions in these epitope regions are known to have a dominant effect on the antigenic evolution of IAV H3N2[2,25]. Epitopes A and B were statistically significantly enriched among the identified 30 sites ($P < 0.05$), which supports previous findings that epitopes A and B are the most immunodominant[26,28].

Seasonal analysis (Fig. 4b) revealed nine sites within epitopes that were consistently ranked in the top 20 sites over all 14 seasons from 2014NH to 2020SH. These comprised three sites in epitope A (140, 144, 145), three in B (158, 186, 189), and three in D (173, 208, 213). The relative importance of epitopes A and B persisted across all seasons, though epitope A appears to have become more important more recently (Fig. 4b, c). Outside epitopes A and B, the importance of epitope D was also reasonably stable across all seasons and this epitope was relatively more important than epitopes C and E (Fig. 4b, c). In earlier seasons (2014NH and 2014SH), site 189 of epitope B was predicted to be the most important antigenic site by our model. This site was previously identified experimentally to be responsible for two antigenic cluster transitions[26] (EN72 to VI75 and BK79 to SI87). This site was also predicted to be the most important antigenic site by NextFlu[7] on a dataset from 2005–2016. In recent seasons (2019SH to 2020SH), our model predicts site 159 of epitope B to be the most important antigenic site. The genetic analysis by Crick WIC[12] showed that most of the viruses circulating in these seasons belong to clade 3C.2a1b.2a.2, where one of its characteristic substitutions includes Y159N resulting in loss of glycosylation that affects recognition of epitopes by antibodies[29].

Of the 30 sites identified across seasons as being most important, five did not belong to any known epitope (Fig. 4a). Among these, four sites are in close proximity (with distance between carbon-alpha atoms <8 Å) to the known epitopes: sites 223 and 241 are located close to epitope D, site 269 is located close to epitope E, while site 225 is close to both epitopes A and D (Fig. 4c). Two of these sites, 183 and 225, are part of the functionally important receptor binding sites (RBS)[28]. Site 225 was consistently ranked in the top 20 important sites across all seasons considered (Fig. 4b). Mutations at site 225 can alter the fitness

landscape of epitope B[30], and a mutation at this site was linked to egg-passaging adaptation in isolates circulating from 2019 to 2021[31].

Overall, our model identified HA1 sites (predominantly within known epitopes but also some outside) that contributed significantly to the antigenic evolution of IAV H3N2 in the last decade and characterized the dynamics of these antigenic "drivers" over time.

## Discussion

We have presented a machine learning model (Fig. 1b, c) that can accurately predict antigenic properties (in terms of NHTs) of IAV H3N2 isolates circulating in an influenza season using only their genetic sequence data and associated metadata. The model was trained and tested under a seasonal framework (Fig. 1a), mimicking the periodic influenza surveillance process followed by WHO for annual vaccine strain selection (Supplementary Fig. 1). The model remained robust under data-limited scenarios.

Computational methods have been developed previously for antigenic characterization of IAV. These include the well-known antigenic cartography[2] method, a multi-dimensional scaling approach that is helpful to visualize and study the relationship among virus isolates and antisera in two dimensions. Other sequence-based models have also been developed[1,6], most of which considered a non-seasonal framework[4,7–11], distinct from the seasonal framework (Fig. 1a) adopted in this work. The non-seasonal framework disregards season/time information and randomly distributes HI titers (or virus isolates) in the multi-seasonal HI data among training and test datasets. Under this framework, the testing data may comprise isolates having antigenic changes that the model has already learned during training, which can lead to overfitting and inflate model performance. In addition to sequence data, information such as structural and physicochemical properties of HA have also been used for IAV antigenic prediction[10,11].

Antigenic changes in influenza HA have been shown to be non-linearly related to genetic changes in recent experimental studies[15,16]. These studies demonstrated that epistatic interactions or specific HA backgrounds can affect the antigenicity of HA substitutions. Thus, linear or additive models that assume independent effects of HA substitutions on antigenicity might be suboptimal for capturing the genetic-to-antigenic relation for HA. By adopting a data-driven ML approach, tree-based models (including AdaBoost, as well as XGBoost and RF) capture nonlinearities in the mapping between genetic and

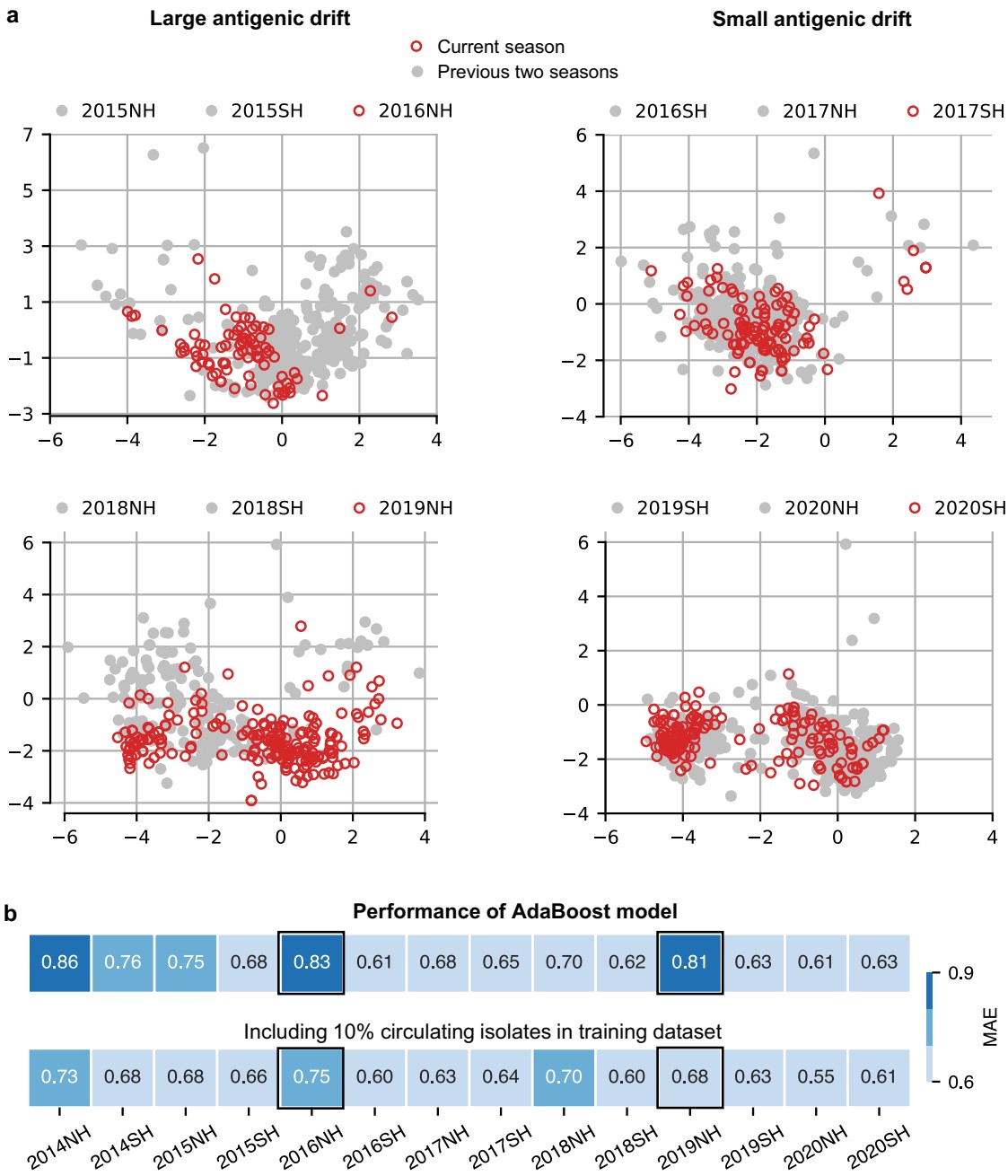

**Fig. 3 | Partial antigenic information mitigates effects of seasonal antigenic drift. a** Antigenic maps[56] to visualize the antigenic drift in circulating isolates compared to isolates from the previous two recent seasons (see "Methods"). The maps on the left show two instances of large antigenic drift in the 2016NH (top-left) and 2019NH (bottom-left) seasons, while the maps on the right show two instances of small antigenic drift in the 2017SH (top-right) and 2020SH (bottom-right) seasons. Each square in a grid indicates the antigenic difference of two units, corresponding to a four-fold dilution of the antibody in the HI assay.

Large antigenic drift is indicated by the presence of circulating isolates (red circles) dispersed far from past isolates (gray points). **b** The MAE performance of the model was evaluated over 14 test seasons, ranging from 2014NH to 2020SH. The top panel displays the MAE performance of the model trained on data from 2003NH up to the corresponding test season. The bottom panel shows the MAE performance of the model when data of randomly selected 10% circulating isolates was included in model training. For each test season, average scores of 50 Monte Carlo runs are reported. Source data are provided as a Source Data file.

antigenic changes. This is shown to yield improved performance when compared to a linear prediction model[7] (Supplementary Fig. 6). Moreover, improved performance is still observed even when the AdaBoost model parameters are matched to those of the linear model (Supplementary Fig. 6). Capturing nonlinearities is however not the only factor which determines the performance, as highlighted by the inferior performance of the nonlinear NN models (MLP and ResNet). This discrepancy in the performance of NN models could be attributed to the tabular structure of the dataset used. Similar findings have been

reported in the literature, indicating diminished performance of NN models when applied to tabular datasets[32].

Previous studies[7,11,33] have demonstrated the value of incorporating virus avidity and antiserum potency in the computational antigenic characterization of IAV H3N2. Our findings highlight the importance of using passage history categories of virus isolates and antisera (e.g., if ferret antisera were raised to cell-propagated or egg-propagated virus isolates), as additional metadata features in model development. Using passage categories alone leads to performance improvement similar to

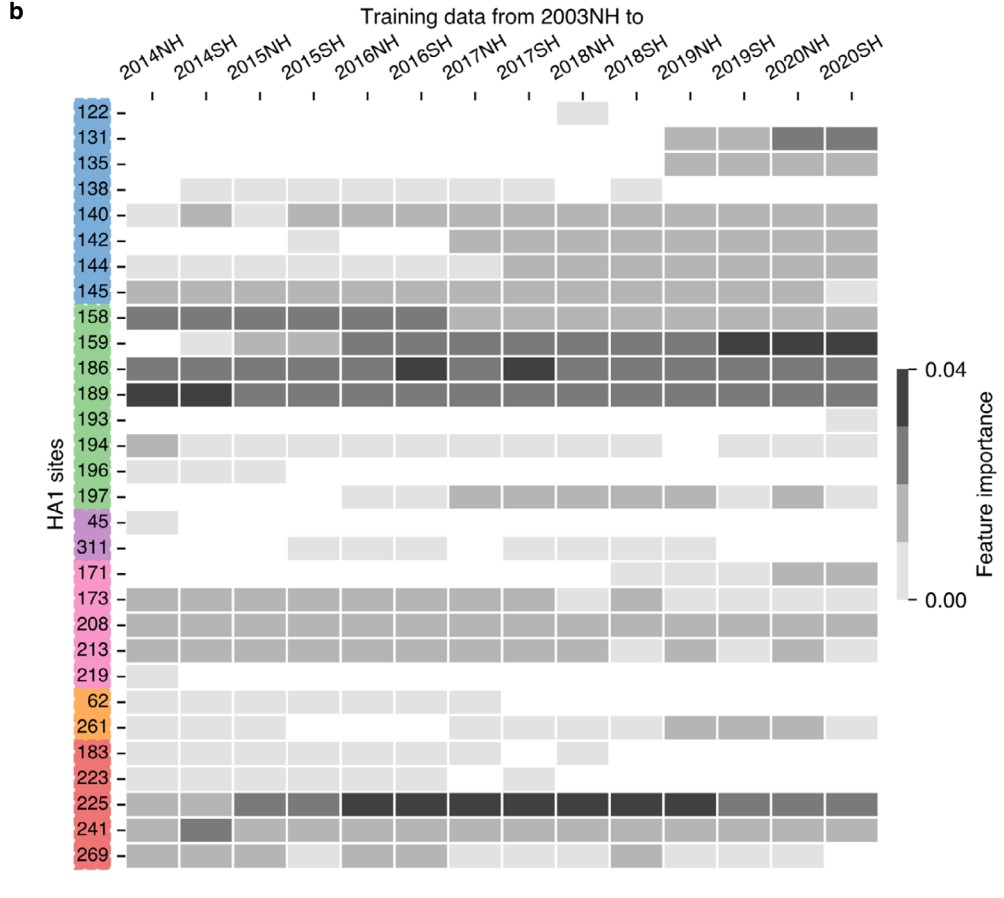

**a**

| Epitope | Important HA1 sites aggregated across seasons |
|---------|-----------------------------------------------|
| A[*]    | 122, 131, 135, 138, 140, 142, 144, 145 |
| B[*]    | 158, 159, 186, 189, 193, 194, 196, 197 |
| C       | 45, 311 |
| D       | 171, 173, 208, 213, 219 |
| E       | 62, 261 |
| Unknown | 183, 223, 225, 241, 269 |

[*] P value < 0.05

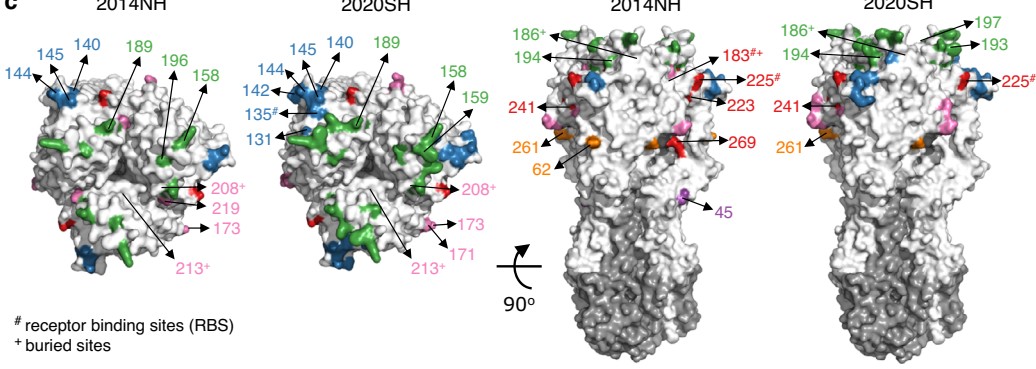

**c**

# receptor binding sites (RBS)
+ buried sites

that of using virus avidity and antiserum potency, and we show that incorporating all of these features together leads to significantly improved model performance (Supplementary Fig. 3a).

The model's predictive power is robust to variations in the training data (Supplementary Fig. 4a, b). Omitting data from a complete season degrades model performance in the following two seasons, but not beyond that (Supplementary Fig. 4c). This indicates that errors due

to a lack of data in specific seasons are not retained in later seasons, and only affect the model's accuracy for a maximum of one or two test seasons. Additional tests showed that training with data from only the two most recent seasons performed similarly to training based on all historical seasons (Supplementary Fig. 8). This is in line with the observed rate of antigenic drift of 1.2 units per year[2] (equivalent to two seasons) for IAV H3N2, which infers that the antigenicity of H3N2

**Fig. 4 | Model-inferred important sites: Correspondence with known epitopes and seasonal dynamics. a** Majority of the 30 important sites identified by the model based on the feature importance scores lie in known IAV H3N2 epitopes. The sites are color-coded according to epitopes. The sites that do not lie in any known epitope are referred to as unknown. *P* value indicates the one-sided statistical significance of epitope enrichment within the identified important sites (see "Methods"). **b** The AdaBoost-based feature importance scores for the 30 important sites are analyzed across subsets of training data from 2003NH to x (x ranges from 2014NH to 2020SH), with the top 20 sites based on the feature importance scores listed for each subset. The darker color cells indicate a higher importance score of a site. **c** Change in the set of important sites, color-coded by epitopes, across two seasons (2014NH and 2020SH) is displayed over the HA structure (Protein Data Bank ID: [6AOU]; A/Brisbane/10/2007). The sites in epitopes A and D are labeled in the top-view (left panel) while the sites in epitopes C, E, and the unknown region are labeled in the front-view (right panel). For epitope B, sites 158, 159, 189, and 196 are labeled in the top-view (left panel) and sites 186, 193, 194, and 197 are labeled in the front-view (right panel). HA1 subunit is shown in white and the HA2 subunit is colored gray. Source data and exact *P*-values are provided as a Source Data file.

isolates would differ substantially beyond two seasons and thereby the corresponding data would likely contribute less to predicting antigenicity of the isolates in the current season.

Some clades of H3N2, e.g., 3C.2a, failed to react in HI assays in the past as they had lost the ability to agglutinate red blood cells (RBC)[34]. To avoid such issues, HI assays are complemented with virus neutralization assays[3]. In comparison to HI assay data, neutralization assay-based antigenic data has been rarely used[5] for developing computational models. This is because the HI assay is still considered the gold standard for characterizing IAV antigenicity, given its well-established protocols and high level of reproducibility and reliability, and in the last few years very few H3N2 viruses do not bind avian or mammalian RBC. Nonetheless, our model can be adapted to predict neutralization titers, a worthwhile problem to pursue in a future study.

To predict NHTs, we used genetic information from the HA1 subunit of the HA protein since it contains the key antibody binding sites (epitopes)[1,2]. Recent research has shown high rates of amino acid substitutions outside the HA1 epitope region as well as in the other influenza surface protein (neuraminidase, NA), possibly indicating positive selection by host immunity[1]. The proposed model could be augmented with genetic information of the HA2 subunit and NA protein for potentially improving prediction accuracy and for examining the role (and temporal dynamics) of HA2 and NA in driving further antigenic changes in IAV H3N2. We focused on IAV H3N2 due to the availability of rich HI titers data for this subtype, as compared to other human influenza viruses[1,21]. We have also adapted the proposed model for IAV H1N1 using a dataset[33,35] spanning 18 influenza seasons from 2001NH to 2009SH. This data set lacked comprehensive passage information, and hence, passage metadata was excluded. We found that the H1N1-adapted model performed seasonal antigenic characterization with an average MAE of 0.747 over these 18 influenza seasons (Supplementary Fig. 9). Our findings motivate the development of methods to track the antigenic evolution of other influenza subtypes, such as IAV (H1N1) pdm09 or influenza B viruses, provided sufficient antigenic data is available.

Serological data is useful not only for guiding IAV vaccine strain selection during VCMs, but also for building computational models addressing general questions related to influenza evolution. These include models for identifying antigenic clusters[2,36,37], and predicting relative growth of viral clades (genetically related isolates stemming from a common ancestor) and forecasting the clades that will likely proliferate in the next season[5,38–41]. The predictions produced by our model can augment experimentally available serological data and can, in turn, be incorporated into models of influenza evolution that use antigenic data.

The AdaBoost model designed for seasonal antigenic characterization has several limitations. First, like any ML model, its performance is contingent on the availability of training data (Supplementary Fig. 2b). Given the limited training data available before 2012 (Supplementary Fig. 2a), we evaluated the model's performance for seasons after 2012 only. Second, regardless of the amount of training data, the model's performance is reduced in seasons with large antigenic drift (Fig. 3a). A small amount of antigenic information of circulating virus isolates (e.g., as low as 10% of the available data) for model training can

help to largely overcome this issue (Fig. 3b). Third, since the model uses amino acid sites as features, it can determine the importance of individual sites (Fig. 4), but not specific amino acid substitutions. This may be addressed in future work by using amino acid substitutions as features for the AdaBoost model. Lastly, while the AdaBoost model can learn a nonlinear genotype-to-phenotype mapping and identify important sites individually, it cannot explicitly identify the collective effects of sites (i.e., epistasis) on antigenicity. Interpretable artificial intelligence techniques, such as Shapley Additive exPlanations (SHAP)[42], may potentially be explored to study the effect of interactions between sites.

Seasonal influenza poses a significant threat to global public health, with high mortality and morbidity rates. The virus's ability to evolve and evade population-level immunity developed from past infections and vaccinations underscores the importance of continued antigenic surveillance for controlling future influenza outbreaks. Noting that only a subset of circulating viruses is tested with HI assays due to practical constraints (e.g., animal availability, resources, cost), our approach could be used to provide normalized HI titers estimates for all sequenced circulating viruses in a given season. This would provide a more comprehensive picture of the antigenic landscape of viruses circulating in each season and could provide complementary input when making vaccine strain selection decisions. Furthermore, our approach can be applied to make rapid sequence-based predictions that suggest which subset of circulating viruses should be tested experimentally with HI assays in a given season. ML-based models, like the one proposed in this work, offer powerful tools for complementing existing antigenic characterization efforts, enabling comprehensive global influenza antigenicity monitoring, improved vaccine strain selection, and effective public health management.

## Methods
### Antigenic and genetic datasets of IAV H3N2
We obtained the antigenic HI titers data for IAV H3N2 from 35 biannual reports published during 2003 – 2021 by the Worldwide Influenza Center (WIC) at the Francis Crick Institute, London[12] (Supplementary data 1). A total of 82,776 HI titers values against virus-antiserum pairs were extracted from these reports, where in each pair the virus represents the circulating/test virus isolate and the antiserum represents the reference virus isolate against which the post-infection ferret antiserum was raised. From these reports, we also extracted the metadata information of virus isolates including their names, passages, and collection dates. Based on the passage information, we labeled each virus isolate with either a cell or egg passage category. We used both the name and passage to represent a unique virus isolate. Invalid HI titers[43] and HI titers of virus-antiserum pairs with passage categories other than egg or cell were removed. Following standard practices of the WHO[2,7], we computed NHT-based antigenic differences for each virus-antiserum pair from the compiled HI titers values. NHT is defined as the difference of the 2-fold dilutions of the homologous and heterologous titers values as follows[2,7]

$$d_{ab} = \log_2\left(T_{b\beta}\right) - \log_2\left(T_{a\beta}\right),$$

where the homologous titers $T_{b\beta}$ and the heterologous titers $T_{a\beta}$ represent the reciprocal of the maximum dilution of antiserum $\beta$ that is required to inhibit cell agglutination by the reference virus isolate $b$ and the test virus isolate $a$, respectively. In case the homologous titers were unavailable, we used the maximum titers value available for that antiserum[2]. We removed the virus-antiserum pairs against which sequences were not found in the influenza genetic databases and used the remaining antigenic data for seasonal antigenic characterization. This included a total of 36,709 NHTs corresponding to 3737 virus isolates paired with 268 antisera.

In addition to NHT, Archetti Horsfall Titers (AHT)[9,44] is also used to characterize antigenic differences between virus isolates. AHT measurement is a two-way analysis[45] that requires four HI assays, and antiserum must be raised against each virus isolate in a pair. AHT is not used by WHO[46] and thus was not considered in this work. We also note that HI assays are dependent on the agglutination of red blood cells. The source of these red blood cells has varied from chicken to turkey and then to guinea pig over the course of time, due to changes in receptor binding sites[47] of IAV H3N2. While these variations are present in the dataset that we consider, the insensitivity of the model to these variations shows that they are likely taken care of by the model parameters of virus avidities and antiserum potencies[7].

For the virus isolates and antisera in this data, we downloaded the corresponding HA protein sequences from the GISAID[13] and the IVR[14] databases. We aligned the HA protein sequences using MAFFT[48] with the full-length HA protein (566 amino acids) of A/Beijing/32/1992 (isolate ID: AAA87553)[7] as a reference. We restricted our model to the HA1 subunit (amino acid sites 17–345) of the HA protein, as this subunit forms the globular head of the HA protein containing key epitopes known to be important for antigenicity of IAV H3N2[1,2].

## Encoding genetic and metadata information

To provide inputs in numeric form to the AdaBoost[17] model, we encoded the genetic sequences of virus isolates using the amino acid mutation matrices in the AAindex2 database[19] (Fig. 1c). As alternatives, we also explored binary and one-hot encoding methods for encoding the genetic sequences of virus isolates (see below for details). The metadata information of virus isolates was represented using the standard one-hot encoding (Fig. 1c). The encoded genetic and metadata information was used as input features of the AdaBoost model.

The AAindex2 database contains 94 $20 \times 20$ amino acid mutation matrices, where each numeric entry of a matrix describes the rate at which an amino acid in a protein sequence is replaced by another amino acid over evolutionary time. These numerical values are based on the physiochemical and biochemical properties of pairs of amino acids. Of these 94 matrices, two matrices (MEHP950101 and MEHP950103) were discarded for being incomplete as they included gaps in their entries. Thus, the remaining 92 matrices were investigated for encoding the genetic information of isolates. Specifically, for each virus-antiserum pair, we computed genetic difference from a reference (antiserum) to test virus isolate by encoding the amino acid mutations at each site of their HA1 protein sequences using the numeric entry of the corresponding amino acid pairs in mutation matrices as described in ref. 44. Briefly, for a specific mutation matrix M, the encoded genetic difference $g$ between the sequences of a virus $v$ and an antiserum $a$ at HA1 position $i$ is given by:

$$g_i = m_{v_i,v_i} + m_{a_i,a_i} - 2m_{a_i,v_i},$$

where $v_i$ and $a_i$ are respectively the amino acids at position $i$ in the virus and antiserum sequence, and $m_{j,k}$ is the entry of the matrix M corresponding to amino acids $j$ and $k$.

In the binary encoding scheme, for each virus-antiserum pair, the amino acid differences at each HA1 site were encoded as '1' and otherwise '0'. Any ambiguous amino acid or gap in the protein

sequences was also encoded as zero to avoid mapping ambiguous genetic information to antigenicity. For each virus-antiserum pair, the binary encoded genetic difference was represented by a binary vector of length 329 corresponding to the length of the HA1 protein sequence.

In the one-hot encoding scheme, for each virus-antiserum pair, at each HA1 site, the amino acids in the two sequences were initially represented as binary vectors of length 20 (corresponding to 20 valid amino acids), as per standard one-hot encoding. Subsequently, a logical OR operation was applied between these two vectors to encode the amino acid differences. Consequently, at each HA1 site, distinct amino acids in a virus-antiserum pair are encoded as a binary vector of length 20 with a pair of ones, each representing a one-hot encoded amino acid. For the alternative case in which the amino acids are the same at a given site, these are encoded into a binary vector of length 20 with a single one, preserving the amino acid information. With this one-hot encoding strategy together with logical OR combining, the genetic difference for each virus-antiserum pair produces an encoded binary vector of length $20 \times 329$.

Each metadata information of isolates—including their virus avidities, antiserum potencies, and passage categories—was considered as categorical data and converted to numeric data using one-hot encoding scheme in Scikit-learn[49]. The encoded vector corresponding to each virus-antiserum pair represents virus avidity, antiserum potency, and passage categories of virus and antiserum. The virus avidity of an isolate is represented by a binary sparse vector of length equal to the number of unique virus isolates in the training dataset, wherein all entries are '0' except a '1' at the position of that isolate in an array of all the virus isolates sorted by their collection dates, names, and then passages. A similar procedure was followed to represent the antiserum potencies corresponding to antisera. For instance, if the training dataset contains 100 unique virus isolates and 10 unique antisera and considering the two passage categories (cell/egg) for isolates corresponding to both virus and antiserum, the one-hot encoding corresponding to each virus-antiserum pair will result in a binary vector of length $100 + 10 + 2 + 2 = 114$. Hence, the one-hot encoding scheme resulted in a sparse binary vector of length equal to the number of categories in each metadata information for the corresponding virus-antiserum pairs in the training dataset. It is worth noting that when predicting the antigenicity of a circulating virus isolate against an antiserum, the virus avidity is represented by a zero vector. This is because the virus itself is not available during the model's training process under the seasonal framework.

## Training, validation, and test datasets

The compiled dataset consisted of 37 influenza seasons from 2003NH to 2021NH. For each test season $s$, the training dataset includes the NHTs corresponding to past virus-antiserum pairs starting from the earliest season 2003NH to the most recent season $s-1$, while the test dataset includes NHTs of the isolates circulating in the test season $s$ paired with the past antisera. The training and test data for each season are described in Supplementary Fig. 2a.

We selected four seasons from 2012NH to 2013SH as validation seasons, which were used for model optimization. The next 14 seasons from 2014NH to 2020SH were selected as test seasons for model evaluation. This selection was based on the stable performance of a baseline model (explained in the next section) over these seasons. Note that virus-antiserum pairs available for the 2021NH season were very limited (Supplementary Fig. 2a). Thus, this season was excluded from the analysis to allow reliable model evaluation. Unlike prior works[21,46,50,51] that used the entire dataset (including test seasons) for model optimization, our model was optimized solely using the data of past seasons to prevent data leakage issues[52] that could inflate model performance[46].

## AdaBoost model

In the designed AdaBoost[17,18] model, the encoded genetic difference at each site of the HA1 protein sequences was treated as an input feature that nonlinearly contributes toward the computation of the NHT. The remaining features of the AdaBoost model consist of binary identifiers for the virus and antiserum related metadata information, including virus avidity, antiserum potency, and their passage categories (Fig. 1c). The designed AdaBoost model is an ensemble of sequentially trained decision trees employing a boosting technique, where each subsequent decision tree seeks to rectify errors present in the preceding decision tree by assigning more weight to training data samples with large errors. At each splitting node of a decision tree, the candidate set of features is a random subset of the features (including encoded genetic differences at each site of the HA1 protein sequences and one-hot encoded metadata information). The predicted NHT by the AdaBoost model is the sum of the weighted predicted NHTs by an ensemble of decision trees.

The baseline model was based on the AdaBoost model with default hyperparameters within the module *AdaBoostRegressor* in Scikit-learn[49], which consists of the following hyperparameters: *n_estimators* = 50, *learning_rate* = 1.0, *loss* = *linear*, *base_estimator* = *decision tree* with a hyperparameter of *max_depth* = 3. We used a random seed equal to 100 for reproducibility. No metadata information was provided to this model and a binary encoding scheme was used. Hence, in this case, NHTs were predicted based on only the binary encoded genetic difference at each site of the HA1 protein sequences of the virus-antiserum pair.

To optimize the AdaBoost model, we performed hyperparameter optimization independently for each of the 92 amino acid mutation matrices as well as for binary and one-hot encoding. We considered two hyperparameters in the module *AdaBoostRegressor* in Scikit-learn[49] with each hyperparameter optimized over a search space defined as follows: *n_estimators*[49]—ranging from 10 to 1000 in steps of 10; and *learning_rate*[49] ranging from 0.1 to 1.5. We set the *estimator* hyperparameter of the *AdaBoostRegressor* to *DecisionTreeRegressor* with its two hyperparameters optimized over a search space defined as follows: *max_depth*[49]—ranging from 1 to 10000 in steps of 10; and *max_features*[49]— ranging from 0.1 to 1. The values of hyperparameter *learning_rate*[49] and *max_features*[49] were sampled from a uniform distribution, while the rest of the hyperparameters were sampled from a quantized uniform distribution[53]. Bayesian optimization procedure termed as the Tree of Parzen Estimator (TPE)[53] under module *hyperopt*[53] was used to automate the process of hyperparameter optimization over 100 runs on the defined search space. The hyperparameter optimization of the AdaBoost model (with binary encoded genetic data and including all metadata features) significantly improved its performance from MAE of 1.091 to 0.759 (Supplementary Fig. 3b). Further, depending on the choice of the mutation matrix the MAE varied between 0.835 to 0.750 (Supplementary Fig. 3c). This performance variation occurs because each mutation matrix incorporates specific amino acid attributes. The optimized AdaBoost model consisted of genetic difference encoded using the best-performing amino acid mutation matrix, GIAG010101[19], and the hyperparameters were set as follows: *n_estimators* = 230, *learning_rate* = 1.393, *max_depth* = 1860, and *max_features* = 0.394. To ensure reproducibility, we maintained a fixed random state of 100 for each Python package across all simulations.

## Performance metrics

To assess the performance of the developed model in a particular season, we computed the MAE between the measured $d$ and predicted $\hat{d}$ NHTs as

$$\text{MAE}_S = \frac{\sum_{(i,j) \in S} \left| d_{ij} - \hat{d}_{ij} \right|}{\#(S)}$$

Here, $S$ denotes the set of virus-antiserum pairs $(i,j)$ in a season and $\#(S)$ represents the cardinality of the set $S$.

To compute the average performance of the model over $N$ test seasons, we used the weighted average of the $\text{MAE}_{S_n}$ obtained for the season $n$, with the weights equal to the cardinality of the dataset in the season $n$. This is given by

$$\text{Average MAE} = \frac{\sum_{n=1}^{N} \#(S_n) \text{MAE}_{S_n}}{\sum_{n=1}^{N} \#(S_n)}$$

To compute the classification scores, the NHTs were converted to binary labels using a threshold of 2 antigenic units[4,21] (equivalent to 4-folds change in HI titers). Thus, a virus-antiserum pair was classified as either antigenic variant (NHT > 2) and assigned a binary label '1' or antigenically similar (NHT ≤ 2) and assigned a binary label '0'. The ability of the model to classify antigenic variants was then determined using standard classification metrics including accuracy, sensitivity, specificity, MCC, and AUROC. Similar to MAE, the classification performance of the model across seasons was computed using a weighted average. Note that the classification threshold can be chosen to improve the classification performance for either antigenic variants or antigenically similar virus-antiserum pairs, considering the target problem. For example, in the scenarios when both sensitivity and specificity are equally important, the threshold can be optimized to maximize Youden's index (sensitivity + specificity − 1) averaged over the most recent three seasons for a given test season (Supplementary Fig. 5b).

## Alternate models

To benchmark the performance of the AdaBoost model for H3N2 antigenic characterization, we compared it with alternate methods. These included a linear method (NextFlu substitution model[7]), two ML methods (RF[22] and XGBoost[23]), and two neural network methods (MLP[24] and ResNet[24]). The implementation details of these methods are provided below.

Linear prediction model (NextFlu): The NextFlu substitution model, a well-known linear model for antigenic prediction, was employed to benchmark our model's ability to capture nonlinearities in the genetic-to-antigenic mapping. In the original work[7], this model was evaluated under a non-seasonal framework. We adapted its implementation (available at https://github.com/nextstrain/augur/blob/master/augur/titer_model.py) to fit our seasonal framework (Fig. 1a) and input data format. Our adapted version, like the original, did not incorporate passage information, and modeled normalized HI titers (NHT) as a linear combination of genetic difference, virus avidity, and antiserum potency.

Machine learning and neural network models: For the RF model, we used the module *RandomForestRegressor* in Scikit-learn[49]. For the XGBoost model, we used module *XGBRegressor* in XGBoost[23]. For the MLP and ResNet models, we used TensorFlow[54] to implement the architectures used in ref. [24]., where the MLP architecture is defined as

$$\text{MLP}(\mathbf{x}) = \text{Linear}(\text{MLPBlock}(\ldots(\text{MLPBlock}(\mathbf{x}))))$$

$$\text{MLPBlock}(\mathbf{x}) = \text{Dropout}(\text{LeakyReLU}(\text{Linear}(\mathbf{x})))$$

and the ResNet architecture is defined as

$$\text{ResNet}(\mathbf{x}) = \text{Prediction}(\text{ResNetBlock}(\ldots \text{ResNetBlock}(\text{Linear}(\mathbf{x}))))$$

$$\text{ResNetBlock}(\mathbf{x}) = \mathbf{x} + \text{Dropout}(\text{Linear}(\text{Dropout}(\text{ReLU}(\text{BatchNorm}(\mathbf{x})))))$$

$$\text{Prediction}(\mathbf{x}) = \text{Linear}(\text{ReLU}(\text{BatchNorm}(\mathbf{x})))$$

where, $\mathbf{x} = [x_1 \ldots x_j]$ is a feature vector of $j$ features corresponding to a single data sample. Here, 'Linear' indicates a fully connected neural network layer, 'Dropout' layer is used for regularization to reduce overfitting that makes the input equal to zero of a few nodes/neurons in a layer, 'BatchNorm' layer normalizes the outputs such that the mean is close to zero and the standard deviation is close to one, and rectified linear unit 'ReLU' and 'LeakyReLU' indicate nonlinear activation functions defined as

$$\text{ReLU}\left(x_j\right) = \max(0, x_j)$$

$$\text{LeakyReLU}\left(x_j\right) = \begin{cases} x_j & \text{if } x_j \geq 0 \\ \alpha x_j & \text{if } x_j < 0 \end{cases}$$

For a fair performance comparison against the AdaBoost model, we optimized the hyperparameters of these models to minimize the average MAE over four validation seasons (2012NH to 2013SH). For optimization of the RF model, we used 92 mutation matrices in the AAindex2[19] database as well as binary encoding. We found that the selection of mutation matrices had a relatively minor effect on the performance of the model (Supplementary Figs. 3c, 10). Based on this observation, we used nine mutation matrices (obtained by combining the set of top five mutation matrices for AdaBoost (Supplementary Fig. 3c) and RF models (Supplementary Fig. 10) for optimizing the remaining models (XGBoost, MLP, and ResNet). As for the AdaBoost model, a Tree of Parzen Estimator (TPE)-based Bayesian optimization procedure[53] under module *hyperopt*[53] was used to automate the process of hyperparameter optimization over 100 runs for the RF and XGBoost models, and the same procedure under module *optuna*[55] was used over 50 runs for the MLP and ResNet models on their defined search space of hyperparameters (Supplementary Table 1). The optimized models include optimal values of their hyperparameters (Supplementary Table 1) for the top-performing mutation matrix (AZAE970101 for RF, GIAG010101 for XGBoost, WEIL970102 for MLP, and MUET010101 for ResNet) and their performance was then evaluated over 14 test seasons (Supplementary Fig. 6).

### Antigenic cartography
To observe antigenic drift of IAV H3N2 isolates across seasons (Fig. 3a), we performed antigenic cartography of these isolates using R's (version 4.2.0) *Racmacs* package[56] (version 1.1.35). *Racmacs* uses the multidimensional scaling procedure, proposed in ref. 2., to position virus isolates and antisera on a lower-dimensional space (2D in our case) based on their HI titers. The 2D coordinates of virus isolates were obtained using default settings of *Racmacs* with 1000 optimizations and setting parameter *minimum_column_basis* to 'none'.

### Feature importance scores of the AdaBoost model
In the AdaBoost model, the feature importance scores depend on the base estimator, which is a decision tree in the proposed model. First, the importance of a feature in each decision tree is determined by how much that feature contributes to increasing leaf purity through variance reduction[22]. The importance scores from each tree are subjected to a weighted average calculation and normalized to a sum of one. The relatively high scores indicate more important features. We computed feature importance scores for all HA1 sites in the proposed AdaBoost model using the built-in function *feature_importances_* in Scikit-learn[49]. To compute these scores, the AdaBoost model was trained on subsets of training data from 2003NH to $x$ ($x$ ranges from 2014NH to 2020SH). For each subset, out of the 329 HA1 sites, we selected the top 20 sites corresponding to the highest feature importance scores.

### Statistical significance of epitope enrichment in top sites (P-values)
Epitope enrichment in the 30 important sites, identified using feature importance scores across seasons (Fig. 4), was calculated using a P value. It represents the probability of observing at least $i$ sites out of $j$ epitope sites in the set of important sites, where the set of important sites comprises 30 sites out of a total of 329 HA1 sites. Mathematically, this can be written as

$$P = \sum_{q=i}^{\min(j,n)} \frac{\binom{j}{q}\binom{329-j}{30-q}}{\binom{329}{30}}$$

The null hypothesis that $i$ epitope sites were observed in the 30 important sites by a random chance was rejected if $P < 0.05$.

### Structural analysis
We used *Pymol* (www.pymol.org) for representing the identified important sites over the three-dimensional HA structure of IAV H3N2 A/Brisbane/10/2007 (available in the Protein Data Bank; PDB ID: [6AOU]). To calculate the distance between an epitope and an identified important site that did not lie in any known epitope, we measured the 3D distance between the carbon-alpha of each epitope site and that of the identified site. The identified site was considered close to the epitope if the calculated distance was less than eight Angstroms for at least one of the epitope's sites.

### Web application
Using streamlit (https://streamlit.io), we have developed a web application that provides an easy-to-use GUI for applying our model to perform seasonal antigenic prediction for IAV H3N2. With this web application, users can directly input full-length (329 amino acids) HA1 sequences of test virus-antiserum pairs and corresponding (optional) metadata information, or they may choose to upload the same data for multiple virus-antiserum pairs using a CSV file. Based on the season of the virus isolates being tested, the web application allows the user to select the appropriate model trained up to (but excluding) the test season.

### Reporting summary
Further information on research design is available in the Nature Portfolio Reporting Summary linked to this article.

## Data availability
The antigenic HI titers data for IAV H3N2 were obtained from biannual reports published by the Worldwide Influenza Center at the Francis Crick Institute, London[12]. The antigenic HI titers data for IAV H1N1 were obtained from the published dataset[35]. The corresponding HA protein sequences for IAV H3N2 and H1N1 were downloaded from the GISAID[13] and the IVR[14] databases. Supplementary Data 1 provides information on the virus-antiserum pairs of IAV H3N2 used in this analysis. It identifies the specific HI data from the Crick WIC reports[12] and the HA protein sequence data from the GISAID[13] and the IVR[14] databases. The three-dimensional HA structure of IAV H3N2 A/Brisbane/10/2007 (PDB ID: [6AOU]) used in this analysis was obtained from the Protein Data Bank (https://www.rcsb.org). The amino acid mutation matrices were obtained from AAindex[19] database. All data used in this work is publicly available as of the date of publication. Source data are provided in this paper.

## Code availability
Source codes implementing the proposed AdaBoost model, and the results presented in this paper can be accessed from GitHub (https://github.com/saws-lab/SAP_H3N2_ML)[57]. The web server running the

web application for seasonal antigenic prediction of IAV H3N2 using our proposed AdaBoost model can be accessed from Hugging Face Spaces (https://huggingface.co/spaces/sawshah/SAP_H3N2). All statistical analyses in this work were performed using Python 3.8.12.

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

## Acknowledgements

The authors thank all of the World Health Organization National Influenza Centers, comprising the WHO Global Influenza Surveillance and Response System (GISRS), for their continuous monitoring of influenza strains around the world. We especially acknowledge the Worldwide Influenza Center at the Francis Crick Institute, London, for sharing influenza antigenic data through reports on their webpage[12], without which this research would not have been possible. We acknowledge all researchers at the originating and submitting laboratories that sequenced influenza viruses and made them available in GISAID and IVR databases. S.A.W.S and D.P.P. were supported by the General Research Fund (16201620) of the Hong Kong Research Grants Council. L.L.M.P. was supported by the Theme-Based Research Scheme (T11-712/19-N) of the Hong Kong Research Grants Council. A.A.Q. and M.R.M. were supported by the Australian Research Council through the Discovery Project (DP230102850). M.R.M. is the recipient of an Australian Research Council Future Fellowship (FT200100928) funded by the Australian Government.

## Author contributions

A.A.Q. and M.R.M. conceived and designed the research. S.A.W.S. curated the datasets and developed the models. All authors contributed to data analysis and data visualization. L.L.M.P. and I.B. provided ideas related to influenza evolution, and D.P.P. provided insights on modeling. A.A.Q. and M.R.M. were responsible for project supervision. S.A.W.S. wrote the original manuscript, and D.P.P., I.B., L.L.M.P., A.A.Q., and M.R.M. reviewed and edited it. All authors discussed and approved the manuscript.

## Competing interests

The authors declare no competing interests.
