## [Peer Review File · Nature Communications]

Seasonal antigenic prediction of influenza A H3N2 using machine learningREVIEWER COMMENTS

Reviewer #1 (Remarks to the Author):

The proposed goal of the study is to develop a machine learning model to predict virus phenotype (~hemagglutination inhibition data) from genotype. The random forest model incorporates minimal genetic information (HA1 similarity), and metadata (avidity, potency, passage history) to predict virus-antiserum cross-reactivity. The authors argue that the “seasonal approach” to training the model is novel, prevents overfitting, and when integrated with metadata has resulted in improved model performance. The authors integrate a significant amount of public HI and sequence data to train the model and apply it to “rank” the importance of individual amino acid sites on virus phenotype.

This contribution is well written, the visualization of the data are strong, the validity/approach taken to build the model is robust (perhaps one-hot encoding the amino acid data without the AAindex step could be considered) and this is an important problem. However, the methods applied are standard and not competed against the state-of-the-art in machine learning, the provided “software” is reproducible for a computational biologist but not for a bench-scientist unfamiliar with these analyses (i.e., it has limited utility), the identification of amino acid sites associated with phenotypic change largely restate prior studies, and the improvement/classification performance represents a marginal improvement. There were intriguing avenues and directions mentioned in the introduction and discussion (e.g., epistasis, the role of sites and genes outside of the HA1, the impact of the immunity landscape on the emergence of antigenic variants) but these were not explored and there was limited empirical testing of the model or an extensive empirical application of the model to provide some form of unique insight into a biological problem.

Consequently, I would suggest that the work be revised to consider and compete against alternate approaches, develop some sort of software beyond the jupyter notebooks so that it could be implemented easily, and an empirical test of the work to address a biological problem should be developed.

Specifically, 1) the proposed model is competed – briefly – with a single method that is integrated in nextstrain. The work would be significantly strengthened if alternate methods were applied, so that this contribution could be better assessed relative to other approaches/the state of the art. Specifically, the RF approach is well studied, and understanding what major advancements are made within this study is difficult. See a handful of recent studies here: Hie, et al 2021. *Science*, 371(6526), pp.284-288; Huddleston, et al., 2020. *Elife*, 9, p.e60067; Zeller, et al. *MSphere*, 6(2), pp.e00920-20.

2) Some form of experiment should be conducted, e.g., what would the impact be on vaccine strain selection should this approach be adopted? Does this approach work for H1 data? Does the emergence of antigenic variants change over time with well-matched/poorly matched vaccines? Is it

amino acid identity, or simply the position that is important, e.g., are all amino acid mutations at 189 equally important?

And 3) the value of a machine learning model – in this context – is that it could be easily deployed by someone with some raw HI data – the git repo is nicely curated, but for a bench-scientist the application of this approach would be very difficult. A suggestion would be a better worked example, create a software that is installed via PyPi or conda, and making it generalizable, e.g., can this be adapted for a virus that is not influenza?

Reviewer #2 (Remarks to the Author):

This study developed a random-forest model for predicting the antigenic variation of influenza A(H3N2) viruses using both the HA sequence and meta data in HI experiments including virus avidity, antiserum potency and passage category of virus isolates and antisera. The novelty of the model lies in the integration of meta data in the model. The study was well designed and the manuscript was written clearly. The reviewer has some major concerns about the study.

1 Lots of computational models have been developed to predict the antigenic variation of influenza viruses. It is not enough to demonstrate the superiority of the model by only comparing it to nextflu. The model had an average accuracy of 0.85 which is not high enough comparing to previous studies.

2 The model relies on metadata in the prediction, which significantly limit its usage in applications.

3 The antigenic evolution of the influenza virus is cluster-wise. It is important to consider the virus population when studying the evolution of the virus.

4 As a new framework for predicting antigenic variation of influenza viruses, it is necessary to test its performance in other subtypes such as A(H1N1) and influenza B virus.

RESPONSE TO REVIEWERS' COMMENTS

Seasonal antigenic prediction of influenza A H3N2 using machine learning

(NCOMMS-23-22338-T)

We thank the reviewers for providing constructive comments. In addressing these comments, we feel the paper has been improved substantially. Our point-by-point responses are detailed below. Changes made to the manuscript are indicated in blue.

Reviewer #1 (Remarks to the Author):

The proposed goal of the study is to develop a machine learning model to predict virus phenotype (~hemagglutination inhibition data) from genotype. The random forest model incorporates minimal genetic information (HA1 similarity), and metadata (avidity, potency, passage history) to predict virus-antiserum cross-reactivity. The authors argue that the “seasonal approach” to training the model is novel, prevents overfitting, and when integrated with metadata has resulted in improved model performance. The authors integrate a significant amount of public HI and sequence data to train the model and apply it to “rank” the importance of individual amino acid sites on virus phenotype.

This contribution is well written, the visualization of the data are strong, the validity/approach taken to build the model is robust (perhaps one-hot encoding the amino acid data without the AAindex step could be considered) and this is an important problem. However, the methods applied are standard and not competed against the state-of-the-art in machine learning, the provided “software” is reproducible for a computational biologist but not for a bench-scientist unfamiliar with these analyses (i.e., it has limited utility), the identification of amino acid sites associated with phenotypic change largely restate prior studies, and the improvement/classification performance represents a marginal improvement. There were intriguing avenues and directions mentioned in the introduction and discussion (e.g., epistasis, the role of sites and genes outside of the HA1, the impact of the immunity landscape on the emergence of antigenic variants) but these were not explored and there was limited empirical testing of the model or an extensive empirical application of the model to provide some form of unique insight into a biological problem.

Consequently, I would suggest that the work be revised to consider and compete against alternate approaches, develop some sort of software beyond the Jupyter notebooks so that it could be implemented easily, and an empirical test of the work to address a biological problem should be developed.

Response:

Thank you for the comments and suggestions. We have now made revisions to the manuscript to address these points. We outline these changes in the following, where we respond to the corresponding, albeit more detailed, review comments.

One specific comment that we address up front (since it is not covered in the comments below), is the use of one-hot encoding to encode the genetic differences between sequences of a virus-antiserum pair. We agree that this is a useful further comparison. Hence, we have now implemented the one-hot encoding scheme into the proposed model. In the revised manuscript, we have incorporated this into Supp. Fig. 3c and updated its caption. We found that in comparison to other genetic encoding schemes, this scheme provided the fourth-best average MAE over four validation seasons (Supp. Fig. 3c). This encoding scheme is explained in updated text in subsection ‘Encoding genetic and metadata information’ under section ‘Methods’, which reads as:

"In the one-hot encoding scheme, for each virus-antiserum pair, at each HA1 site the amino acids in the two sequences were initially represented as binary vectors of length 20 (corresponding to 20 valid amino acids), as per standard one-hot encoding. Subsequently, a logical OR operation was applied between these two vectors to encode the amino acid differences. Consequently, at each HA1 site, distinct amino acids in a virus-antiserum pair are encoded as a binary vector of length 20 with a pair of ones, each representing a one-hot encoded amino acid. For the alternative case in which the amino acids are the same at a given site, these are encoded into a binary vector of length 20 with a single one, preserving the amino acid information. With this one-hot encoding strategy together with logical OR combining, the genetic difference for each virus-antiserum pair produces an encoded binary vector of length 20×329 ."

For the remaining review comments, our point-by-point responses are detailed below.

Specifically,

- 1) *the proposed model is competed – briefly – with a single method that is integrated in nextstrain. The work would be significantly strengthened if alternate methods were applied, so that this contribution could be better assessed relative to other approaches/the state of the art. Specifically, the RF approach is well studied, and understanding what major advancements are made within this study is difficult. See a handful of recent studies here: Hie, et al 2021. Science, 371(6526), pp.284-288; Huddleston, et al., 2020. Elife, 9, p.e60067; Zeller, et al. MSphere, 6(2), pp.e00920-20.*

Response:

Thank you for the constructive comment. The novelty of our results lies in the demonstration of accurate *seasonal* antigenic prediction protocols for influenza, which has not been shown previously. This is practically important and relevant to periodic influenza surveillance and vaccine selection. Numerous alternative predictive algorithms to a random forest (RF) approach can nonetheless also be adopted in our seasonal framework. In the original manuscript, we provided a comparison with NextFlu (implemented in our seasonal framework) since it is a widely known method and is incorporated into the NextStrain prediction framework. It is also a linear method, and hence, comparison with NextFlu enabled us to assess the potential advantages of relaxing a linear constraint by adopting a nonlinear RF approach.

We have now implemented and evaluated multiple machine learning (ML) methods in our seasonal prediction framework; see **Figure R1** (Supp. Fig. 6 in the revised manuscript). Specifically, along with the RF and NextFlu algorithms considered earlier, we have implemented two additional tree-based learning methods, adaptive boosting (AdaBoost) (also used in the quoted reference (Zeller *et al.*, 2021), albeit in a non-seasonal framework) and extreme gradient boosting (XGBoost), as well as two neural network methods, multilayer perceptron (MLP) and residual neural network (ResNet). In all cases, passage-based metadata features (Fig. 1b) were incorporated, and we optimized the hyperparameters with respect to a set of amino acid mutation matrices in the AAindex2 database. With this new analysis, we found that the AdaBoost model performed the best among all methods (**Figure R1**). As such, in the revised manuscript, we have selected AdaBoost as our proposed model for seasonal antigenic prediction of influenza A virus (IAV) H3N2 (rather than RF, considered previously).

For the AdaBoost model, we have re-run all simulations and updated all figures and corresponding results in the paper. The performance results are generally improved compared with our earlier analysis, however, the qualitative results and conclusions remain consistent. The main changes are provided in the revised manuscript in Results subsections 'Machine learning model for seasonal antigenic characterization of IAV H3N2' and 'Optimized model accurately performs seasonal antigenic characterization', Methods subsections 'AdaBoost model', and 'Alternate methods'. See also the updated Figures 1 to 4, Supplementary Figures 2 to 8, the new Supplementary Figures 9 to 10, and Supplementary Table 1.

Figure R1. Performance comparison of ML and NN models for antigenic prediction of IAV H3N2 under the seasonal framework. Comparison of the proposed AdaBoost model with a linear model (NextFlu substitution model), tree-based ML models (RF and XGBoost), and NN models (MLP and ResNet). See *Methods* for implementation details of these models. The "AdaBoost (NextFlu-matched-params)" model is based on AdaBoost, with parameters tailored to match those of the NextFlu model that uses binary-encoded genetic differences and only two metadata features: virus avidity and antiserum potency. For each model, MAE was computed for 14 test seasons from 2014NH to 2020SH.

We have also considered the quoted articles by Hie et al., 2021 and Huddleston et al., 2020. However, the approaches in these articles are not directly applicable to the seasonal antigenic prediction problem that we address. Specifically, the model presented by Hie et al., 2021 defined the semantic change (corresponding to antigenic change) as the L1 norm between the wild-type sequence and a single-residue mutant, and the model is trained in an unsupervised fashion without HI titre data. Hence, it cannot predict the outcome of HI assays. The model designed by Huddleston et al., 2020 is an extension of the NextFlu method that forecasts the genetic composition of the future season's influenza population. Hence, rather than predicting HI titres, this model predicts the amino acid sequence that will likely dominate in the next influenza season.

- 2) *Some form of experiment should be conducted, e.g., what would the impact be on vaccine strain selection should this approach be adopted? Does this approach work for H1 data? Does the emergence of antigenic variants change over time with well-matched/poorly matched vaccines? Is it amino acid identity, or simply the position that is important, e.g., are all amino acid mutations at 189 equally important?*

Response:

Considering these points, we have now conducted further experiments, as we describe in the following.

Starting with the extension to H1 data, this is indeed possible. While our study is focused on seasonal H3N2, the proposed seasonal antigenic prediction method and framework is general and can be easily applied to other influenza subtypes, provided sufficient data is available for model training and evaluation. To demonstrate this, we have now applied our method to a publicly available data set that has been made available for seasonal IAV H1N1 (Gregory et al., 2016) spanning 18 influenza seasons from 2001NH to 2009SH. We evaluated the performance of our AdaBoost-based seasonal prediction approach on this data set, with results shown in **Figure R2**. Note that this H1N1 data set lacked comprehensive passage information, and hence, passage meta-data was excluded. On average, the H1N1 system achieved a comparable but higher average MAE compared with H3N2 (0.75 versus 0.70). This higher average MAE is likely attributed to the lack of passage information, particularly given that the passage information was found to significantly improve prediction performance for H3N2 (Supp.

Fig. 3a). Seasonal fluctuations in performance were also more pronounced for H1N1, with season 2005SH achieving a significantly low MAE, while seasons 2003NH and 2007NH exhibited relatively high MAEs (**Figure R2**). These fluctuations can be attributed to the observed antigenic drift during those seasons (**Figure R3**). Specifically, seasons 2004SH and 2005SH displayed a minor antigenic drift (**Figure R3b** and **Figure R3c**), while season 2007NH showed a more substantial antigenic drift (**Figure R3d**). In the case of season 2003NH, although the antigenic drift appears to be small (**Figure R3a**), the higher predicted MAE may also be influenced by the limited availability of circulating isolates during that season (Supp. Fig. 9a).

We have included this result in the revised text in the 'Discussion' section and added **Figure R2** as a (new) Supplementary Fig. 9b.

Figure R2. The MAE performance of the proposed AdaBoost model for seasonal antigenic prediction of IAV H1N1 over 20 influenza seasons from 2001NH to 2009SH. The 'Average' cell on the right indicates the score averaged over the 20 seasons. The darker colour cells indicate better performance.

Figure R3. Antigenic maps for IAV H1N1 to visualize the antigenic drift in circulating isolates compared to isolates from the previous two recent seasons. (a-c) Small antigenic drift in seasons 2003NH, 2004SH, and 2005SH. (d) Large antigenic drift in season 2007NH. Each square in a grid indicates antigenic difference of two units, corresponding to a four-fold dilution of the antibody in the HI assay. Large antigenic drift is indicated by presence of circulating isolates (red circles) dispersed far from past isolates (grey points).

Regarding the dependence of antigenic variant changes on vaccine match, we agree that this is an interesting question to explore. In our original manuscript, with the help of antigenic cartography tools, we had demonstrated that the level of seasonal antigenic drift had an effect on the performance of our model (Fig. 3). More specifically, we showed that prediction accuracy was inversely related to the level of drift experienced in a season. Prompted by the reviewer's query, we conducted further analysis to assess whether there was any association between the performance of our prediction method (and consequently, with the level of antigenic drift) and the match of the seasonal IAV H3N2 vaccine strain. Following Qiu et al., 2020, we utilized a simple empirical measure called the 'vaccine similarity' metric to estimate the level of match between the vaccine and circulating viruses. This metric quantifies the percentage of virus-vaccine pairs that exhibit antigenic similarity, indicated by NHT values of two or less. We note that this vaccine similarity metric is only a coarse approximation of vaccine match, and its accuracy is influenced by the number of virus-vaccine pairs reported in a particular season in the WIC reports. The accuracy of this metric may also be affected by potential "selection biases", since the isolates that have been evaluated with HI titres in each season may not represent a completely random sampling of circulating viruses in that season. Nevertheless, using this definition, our analysis revealed a weak negative correlation between prediction accuracy of our approach and vaccine similarity (**Figure R4**). This correlation was however not statistically significant ($P>0.05$). This data also suggests that there is also no clear association between the level of antigenic drift and vaccine similarity. To see this more clearly, we refer to the antigenic cartography plots given in Fig. 3 of our manuscript, which show that the 2016NH and 2019NH seasons experience high antigenic drift, while the 2017SH and 2020SH seasons experience low antigenic drift. From **Figure R4**, there is no clear association between the level

Correlation coefficient	virus-vaccine pairs	p value
Pearson	-0.41	0.14
Spearman	-0.46	0.1

Figure R4. Seasonal vaccine similarity versus the seasonal predictive performance of our model. For each season from 2014NH to 2020SH, the vaccine similarity is plotted with a blue line, with its scale shown on the left y-axis. For each season, vaccine similarity is defined as the percentage of antigenically similar virus-vaccine pairs based on the normalised HI titre data for that season. For each test season from 2014NH to 2020SH, the MAE performance of the AdaBoost model for virus-vaccine pairs is respectively plotted in orange. The MAE scale is shown on the right y-axis. The Pearson and Spearman correlation coefficients between vaccine similarity and MAE performance are shown in the table. Both results were statistically insignificant.

of antigenic drift and vaccine similarity in these seasons. Overall, our data suggests that the prediction accuracy of our model is related to the level of antigenic drift experienced in an influenza season but is not obviously related to the match of the vaccine strain in each season. The level of drift in a season and the vaccine match also do not appear to show any clear relationship.

Regarding the potential impact of our proposed method on influenza vaccine strain selection, there are various ways in which the proposed prediction framework may be applied. These include:

- *Informing targeted HI experimentation:* Our approach can be applied to make rapid sequence-based predictions that suggest which subset of circulating viruses should be tested experimentally with HI assays in a season.
- *Providing complementary HI data predictions:* Noting that only a subset of circulating viruses is tested with HI assays due to practical constraints (e.g., animal availability, resources, cost), our approach could be used to provide normalized HI titre estimates for *all* sequenced circulating viruses in a season. This would provide a more comprehensive picture of the antigenic landscape of viruses circulating in each season and could provide complementary input when making vaccine strain selection decisions. (Note: These complementary predictions are expected to be particularly accurate, as suggested by Supp. Fig. 7 in the manuscript.)
- *Predicting the efficacy of existing vaccines, or candidate vaccines, in a forthcoming influenza season:* When used together with influenza evolutionary forecasting tools that predict the dominant clade/influenza quasispecies in a forthcoming influenza season, our approach can be used to predict antigenic differences between putative circulating strains in that season and strains used in existing or candidate vaccines. This could provide important input when deciding on the need for a vaccine update and for optimal vaccine strain selection.

We have updated the Discussion section of the manuscript to make these points clear.

Regarding the importance of amino acid substitutions, the amino acid mutation matrix encoding that we employ encodes the positional information of amino acids, however it lacks the resolution to determine the importance of individual amino acid substitutions at specific positions. We point out this limitation of our model in the section, 'Limitations of the study'. We also point out that this limitation could be addressed in the future by including amino acid substitutions as features of the model, which will further require re-optimization of the model such as selection of the top performing metadata features and re-optimization of the model hyperparameters. The updated text is as follows:

Third, since the model uses amino acid sites as features, it can determine the importance of individual sites (Fig. 4), but not specific amino acid substitutions. Hence, our model cannot rank different amino acid substitutions that occur at the same site. This may be addressed in future work by using amino acid substitutions as features for the AdaBoost model.

- 3) *And the value of a machine learning model – in this context – is that it could be easily deployed by someone with some raw HI data – the git repo is nicely curated, but for a bench-scientist the application of this approach would be very difficult. A suggestion would be a better worked example, create a software that is installed via PyPi or conda, and making it generalizable, e.g., can this be adapted for a virus that is not influenza?*

Response:

Thanks for raising this point that will add to the value of this work. For a bench-scientist, we believe that an appropriate and broadly accessible use case would be to provide a system that takes as input one or more test virus-antiserum pairs for an influenza season, and reports the predicted normalised HI titre values for these pairs. To facilitate this, we have developed a user-friendly web application with a

graphical user interface (GUI) (**Figure R5**). This web application allows bench-scientists to (i) directly input sequences of test virus-antiserum pairs and corresponding (optional) metadata information, or (ii) upload the same data for multiple virus-antiserum pairs using a CSV file. Based on the season of the virus isolates being tested, the web application allows the user to select the appropriate model trained up to (but excluding) the test season. To demonstrate its usage, an example and instructions for preparing an input CSV file are also provided. The web application is hosted on the Hugging Face Spaces platform, accessible at https://huggingface.co/spaces/sawshah/SAP_H3N2. This web server eliminates the need for scientists to install and run an application via PyPi or conda, making it more convenient and accessible. We refer to the web application and provide a link to access it in the 'Results' section, while the specific details and information about the web application are explained in the 'Methods' section.

In addition to this web application, the updated codes in the form of Jupyter notebooks are provided over GitHub (https://github.com/saws-lab/SAP_H3N2_ML) and can be cloned via Git. These notebooks currently reproduce the results for IAV H3N2 and IAV H1N1 present in the manuscript. Research scientists can use these notebooks to retrain the model on user's data as well as can modify them for other viruses. For example, for viruses having neutralization assay data such as SARS-CoV-2, dengue virus, and hepatitis C virus, the provided notebooks can be modified to take as input neutralization data instead of HI assay data. This extension would be worth considering in a future study. As the focus of this work is on IAV H3N2 and we have also showed its adaptability for IAV H1N1, we pointed out in the 'Discussion' section of the manuscript that this approach could be used for other viruses with appropriate adaptation.

Select the influenza season of your test virus isolates

2021NH

For a single virus-antiserum pair, select 'Input' option, and for multiple pairs, select 'Upload' option.

Input data or File upload?

Input

Seasonal antigenic prediction of influenza A virus (IAV) H3N2

Predict the NHT-based antigenic difference between virus-antiserum pairs using their HA1 sequences and (optional) metadata information.

Input HA1 sequences and (optional) metadata

HA1 sequence of virus isolate

```

          QKIPGNDNSTATLCLGHAVPNGTIVKTIITNDRIEVTNATELVQNSSIGEICDSPHQILDGGNCTLIDALLGDPQC
          DGFQNKWDLFVERNKAYSNCPYDVPDYASLRSLVASSGTLEFNNEFNFNAGVTQNGTSSACIRGSSSFFSR
          LNWLTHLNNIYPAQNVTPMKNKEQFDKLYIYGVHHPDPTDKNQISLFAQSSGITVSTKRSQQAVIPNIGSRPRIRD
          PSRISIWTVIKPGDILLINSTGNLIAPRGYFKIRSGKSSIMRSDAPIGKCKSECITPNGSIPNDKPFQVNNRITYGAC
          PRVVKQSTLKLATGMRNVPEKQTR
          
```

329/329

HA1 sequence of antiserum/vaccine (reference virus isolate)

```

          QKIPGNDNSTATLCLGHAVPNGTIVKTIITNDRIEVTNATELVQNSSIGEICDSPHQILDGENCTLIDALLGDPQC
          DGFQNKWDLFVERNKAYSNCPYDVPDYASLRSLVASSGTLEFNNEFNFNAGVTQNGTSSACIRGSSSFFSR
          LNWLTHLNSKYPALNVTMPNNEQFDKLYIYGVHHPDPTDKNQISLFAQSSGRITVSTKRSQQAVIPNIGSRPRIRD
          IPSRISIWTVIKPGDILLINSTGNLIAPRGYFKIRSGKSSIMRSDAPIGKCKSECITPNGSIPNDKPFQVNNRITYGA
          CPRVVKQSTLKLATGMRNVPERQTR
          
```

329/329

Name of virus isolate (if unknown, leave it blank)

A/MICHIGAN/173/2020

Name of antiserum/vaccine (if unknown, leave it blank)

A/KANSAS/14/2017

Passage of virus isolate (if unknown, leave it blank)

S1

Passage of antiserum/vaccine (if unknown, leave it blank)

E5

Predict

Figure R5. Snapshot of the user-friendly GUI-based web application for the developed model.

Reviewer #2 (Remarks to the Author):

This study developed a random-forest model for predicting the antigenic variation of influenza A(H3N2) viruses using both the HA sequence and meta data in HI experiments including virus avidity, antiserum potency and passage category of virus isolates and antisera. The novelty of the model lies in the integration of meta data in the model. The study was well designed and the manuscript was written clearly. The reviewer has some major concerns about the study.

- 1) *Lots of computational models have been developed to predict the antigenic variation of influenza viruses. It is not enough to demonstrate the superiority of the model by only comparing it to nextflu. The model had an average accuracy of 0.85 which is not high enough comparing to previous studies.*

Response:

Thank you for the comments. Regarding the novelty of the approach, the incorporation of meta-data (particularly passage information), is indeed a distinction of our method compared with other predictive strategies in the literature. As we show in Section “Model training, optimization, and validation”, this can lead to considerable improvement in prediction accuracy. It is important to emphasize however that the novelty of our manuscript is not restricted to the inclusion of meta-data. The problem that we address in this work – predicting HI titres in an influenza season from sequence and meta data alone – has not been addressed previously. This is a highly relevant problem for influenza surveillance and vaccine strain selection. It seeks to characterise the antigenic distinction of circulating viral strains compared with strains from previous seasons, including vaccine strains. Most previous attempts of predicting HI titres address a different problem, in which HI titres are predicted that are randomly selected in time, and typically, they seek to predict strains in the past, based on future HI data. This is a considerably easier prediction problem compared with predicting the antigenic properties of newly circulating strains when they arise in a new season. Our approach is the first to demonstrate that HI titres can be predicted for circulating strains, on a season-by-season basis, which we believe is a key novel aspect of our work and is important in practice.

With the above in mind, it is true that alternative prediction methods can be substituted and evaluated in the seasonal prediction framework that we consider. Our initial analysis provided a comparison with NextFlu since this is a widely known method and is incorporated into the NextStrain framework. It is also a linear method, and hence, comparison with NextFlu enabled us to assess the potential advantages of relaxing a linear constraint by adopting a nonlinear RF approach. We have now implemented and evaluated multiple machine learning and neural network methods in our seasonal prediction framework; with results reported in **Figure R1** (Supp. Fig. 6 in the revised manuscript). Specifically, along with the RF and NextFlu algorithms considered earlier, we have implemented two additional tree-based learning methods, adaptive boosting (AdaBoost), and extreme gradient boosting (XGBoost), as well as two neural network methods, multilayer perceptron (MLP) and residual neural network (ResNet). In all cases, passage-based metadata features (Fig. 1b) were incorporated, and we optimized the hyperparameters with respect to a set of amino acid mutation matrices in the AAindex2 database. With this new analysis, we found that the AdaBoost model performed the best among all methods (**Figure R1**). As such, in the revised manuscript, we have selected AdaBoost as our proposed model for seasonal antigenic prediction of influenza A virus (IAV) H3N2 (rather than RF, considered previously). For the AdaBoost model, we have re-run all simulations and updated all figures and corresponding results in the paper. The performance results are generally improved compared with our earlier analysis, however the qualitative results and conclusions remain consistent. The main changes are provided in the revised manuscript in Results subsections ‘Machine learning model for seasonal antigenic characterization of IAV H3N2’ and ‘Optimized model accurately performs seasonal antigenic characterization’, Methods subsections ‘AdaBoost model’, and ‘Alternate methods’. See also the updated Figures 1 to 4, Supplementary Figures 2 to 8, the new Supplementary Figures 9 to 10, and Supplementary Table 1.

Figure R1. Performance comparison of ML and NN models for seasonal antigenic prediction of IAV H3N2 under the seasonal framework. Comparison of the proposed AdaBoost model with a linear model (NextFlu substitution model), tree-based ML models (RF and XGBoost), and NN models (MLP and ResNet). See *Methods* for implementation details of these models. The "AdaBoost (NextFlu-matched-params)" model is based on AdaBoost, with parameters tailored to match those of the NextFlu model that uses binary-encoded genetic differences and only two metadata features: virus avidity and antiserum potency. For each model, MAE was computed for 14 test seasons from 2014NH to 2020SH.

Regarding the accuracy of our approach, initially we demonstrated an average MAE of 0.74 (normalized HI titre regression) and AUROC of 92% (antigenic variant classification). Notwithstanding some fluctuations over different seasons, these results overall (Figure 2) show a clear ability of our approach to accurately classify antigenically distinct variants for each influenza season. The MAE of 0.74 also reflects accurate estimation of NHTs, particularly considering that antigenic distinction is routinely considered as NHTs greater than 2 antigenic units. Our newly updated results with AdaBoost show slightly further improved performance also.

We believe that our performance results cannot be meaningfully compared with previous antigenic accuracy results presented in the literature, which deal with a different prediction problem as described in our comments above. We further point out in the Discussion section of our manuscript that for such prior analyses based on a non-seasonal prediction framework, the testing data may comprise isolates having antigenic changes that the model has already learned during training, which can lead to overfitting and inflate model performance. To demonstrate these points more explicitly, we have conducted further analysis using the NextFlu model. Under a non-seasonal prediction framework following that considered in the original NextFlu publication (Neher *et al.*, 2016), the NextFlu model achieved an average MAE of 0.502 (**Figure R6**). This is substantially lower than the MAE achieved with NextFlu when implemented in the seasonal antigenic prediction framework considered in our paper (0.819 MAE; see Supp. Fig. 6). Since these prediction frameworks are very different, we believe it is not meaningful to draw performance comparisons between them.

Figure R6. Under the non-seasonal framework, the prediction accuracy of the adapted NextFlu substitution model (MAE = 0.502) matches with the original model (MAE = 0.5). The NHTs predicted by the NextFlu substitution model (y-axis) correlated well with the measured NHTs (x-axis). The evaluation strategy was set similar to that reported in the original work. Specifically, genetic and antigenic data in the 12-year timeframe 2003 – 2015 corresponding to only cell passage category was considered. The data was then randomly divided into 90% training and 10% test datasets in order to train and evaluate the performance of the model, respectively.

2) *The model relies on metadata in the prediction, which significantly limit its usage in applications.*

Response:

The incorporation of meta-data enhances model prediction, and in practice, we do not expect this to be a limiting factor. The meta-data information encoded in our model, which includes virus avidity, antiserum potency, and passage, simply involves the name and passage information of the two viruses used in each HI assay (i.e., the virus used to generate antiserum and that used for testing). For H3N2, this information is readily available in the Crick Institute WHO reports, and it is routinely reported as part of influenza surveillance based on HI assays.

In terms of *testing* or *applying* our trained model in practice to predict the outcomes of HI assays (and more specifically, normalized HI titres), users would simply provide the HA1 sequences for the two viruses to be used in the HI assay, along with the names of those viruses and associated passaging system used for virus culturing (e.g., egg or cell). For the latter, a user could test different inputs to explore a variety of passaging approaches, or they could input their passage of choice (e.g., based on what passaging system may be available to them or their lab in practice). Our prediction system will also function, albeit with performance degradation, if no meta-data information is provided (see Supp. Fig. 3(a)). This flexibility is further highlighted in a web-based application that we have now developed, which seeks to enable scientists (including wet-lab scientists) to make HI assay predictions using our trained algorithm. This system asks users to input a pair of HA1 sequences for the virus isolate and antiserum, and *optionally* the names and passaging information of both. The system will then report the predicted normalized HI values. Based on the season of the virus isolates being tested, the web application allows the user to select the appropriate model trained up to (but excluding) the test season. The web application is hosted on the Hugging Face Spaces platform, accessible at https://huggingface.co/spaces/sawshah/SAP_H3N2.

While not an issue for H3N2, for any specific data sets where complete meta-data information is not available for model training, the approach is also easily adaptable. This is demonstrated in a model that we have now implemented based on a H1N1 dataset (see the response to the Comment 4 below), for which comprehensive passage information was not available.

3) *The antigenic evolution of the influenza virus is cluster-wise. It is important to consider the virus population when studying the evolution of the virus.*

Response:

The antigenic evolution of H3N2 has indeed been shown in prior studies to undergo some level of punctuated, or cluster-wise, antigenic evolution. Most notable is the seminal work by Smith et al., 2004, which reported 10 antigenic cluster transitions over the span of 35 years, where on average a cluster remained dominant for 3.3 years. Cluster transitions are related to large antigenic drift of virus isolates in a season with respect to virus isolates from previous seasons. The results of Smith et al., 2004 demonstrate that cluster transitions do not occur frequently, and for most influenza seasons the antigenic evolution is more mild. While to our knowledge antigenic clusters have not been clearly reported in more recent influenza seasons, analysis of our dataset shows a few specific seasons for which large antigenic drift occurred; namely, the 2016NH and 2019NH seasons (Fig. 3a) between 2016 to 2020.

As far as our predictive model is concerned, if a cluster transition were to occur in a season, the system would simply make prediction for circulating viruses with relatively large antigenic drift. A characteristic feature of our model is that it adapts to new data as it becomes available (since it is progressively trained each season as new antigenic data becomes available), with the past two seasons carrying the most predictive power (see Supp. Fig. 8). This ensures that the model is continually updated in line with the antigenic evolution of influenza, regardless of whether a large (cluster defining) antigenic change or more mild antigenic drift occurs in individual seasons. Moreover, by evaluating performance for every season from 2014NH to 2020SH, we see that the model gives robust predictions in all cases, irrespective of whether a putative cluster-transition event occurs or not. For the two seasons where large antigenic drift most clearly occurs (2016NH and 2019NH), our model performance degrades slightly, but we show that this can be greatly improved with partial HI titre data in the season (see Fig. 3b).

Hence, while our model does not explicitly incorporate cluster-transition information, it does capture and adapt to cluster-transition events and continues to provide robust predictions.

4) *As a new framework for predicting antigenic variation of influenza viruses, it is necessary to test its performance in other subtypes such as A(H1N1) and influenza B virus*

Response:

Thank you for the suggestion. While our study is focused on seasonal H3N2, the proposed seasonal antigenic prediction method and framework is general and can be easily applied to other influenza subtypes, provided sufficient data is available for model training and evaluation. To demonstrate this, we have now applied our method to a publicly available data set that has been made available for seasonal IAV H1N1 (Gregory et al., 2016) spanning 18 influenza seasons from 2001NH to 2009SH. For this data set, comprehensive passage information was not available, hence only virus avidity and antiserum potency were encoded as meta-data (based on the virus names). We evaluated the performance of our AdaBoost-based seasonable prediction approach on this dataset, with results shown in **Figure R2**. On average, the H1N1 system achieved a comparable but higher average MAE compared with H3N2 (0.75 versus 0.70). This higher average MAE is likely attributed to the lack of passage information, particularly given that the passage information was found to significantly improve prediction performance for H3N2 (Supp. Fig. 3a). Seasonal fluctuations in performance were also more pronounced for H1N1, with season 2005SH achieving a significantly low MAE, while seasons 2003NH and 2007NH exhibited relatively high MAEs (**Figure R2**). These fluctuations can be attributed to the observed antigenic drift during those seasons (**Figure R3**). Specifically, seasons 2004SH and 2005SH displayed a minor antigenic drift (**Figure R3b** and **Figure R3c**), while season 2007NH showed a more

substantial antigenic drift (**Figure R3d**). In the case of season 2003NH, although the antigenic drift appears to be small (**Figure R3a**), the higher predicted MAE may also be influenced by the limited availability of circulating isolates during that season (Supp. Fig. 9a).

We have included this result in the revised text in the 'Discussion' section and added **Figure R2** as a (new) Supplementary Figure 9b.

Currently, there is a lack of available antigenic data for influenza B. Compared with H3N2, the antigenic evolution of influenza B has been less investigated. Once sufficient data for influenza B becomes available, a similar seasonal predictive model for this subtype can be developed.

Figure R1 The MAE performance of the proposed AdaBoost model for seasonal antigenic prediction of IAV H1N1 over 20 influenza seasons from 2001NH to 2009SH. The 'Average' cell on the right indicates the score averaged over the 20 seasons. The darker colour cells indicate better performance.

Figure R3 Antigenic maps for IAV H1N1 to visualize the antigenic drift in circulating isolates compared to isolates from the previous two recent seasons. (a-c) Small antigenic drift in seasons 2003NH, 2004SH, and 2005SH. (d) Large antigenic drift in season 2007NH. Each square in a grid indicates antigenic difference of two units, corresponding to a four-fold dilution of the antibody in the HI assay. Large antigenic drift is indicated by presence of circulating isolates (red circles) dispersed far from past isolates (grey points).

REFERENCES

- Gregory, V. *et al.* (2016) 'Human former seasonal Influenza A (H1N1) haemagglutination inhibition data 1977-2009 from the WHO collaborating centre for reference and research on influenza, London, UK'.
- Hie, B. *et al.* (2021) 'Learning the language of viral evolution and escape', *Science*, 371(6526), pp. 284–288.
- Huddleston, J. *et al.* (2020) 'Integrating genotypes and phenotypes improves long-term forecasts of seasonal influenza A/H3N2 evolution', *Elife*, 9, p. e60067.
- Neher, R.A. *et al.* (2016) 'Prediction, dynamics, and visualization of antigenic phenotypes of seasonal influenza viruses', *Proceedings of the National Academy of Sciences*, 113(12), pp. E1701–E1709.
- Qiu, T. *et al.* (2020) 'A benchmark dataset of protein antigens for antigenicity measurement', *Scientific Data*, 7(1), pp. 1–8.
- Smith, D.J. *et al.* (2004) 'Mapping the antigenic and genetic evolution of influenza virus', *Science*, 305(5682), pp. 371–376.

REVIEWERS' COMMENTS

Reviewer #3 (Remarks to the Author):

In my opinion, the authors have thoroughly addressed both of the previous reviewers' concerns.